# Directional thermal emission and display using pixelated non-imaging micro-optics

Ziwei Fan[1,2], Taeseung Hwang[3], Sam Lin[3], Yixin Chen[3] & Zi Jing Wong [1,2,3] ✉

Thermal radiation is intrinsically broadband, incoherent and non-directional. The ability to beam thermal energy preferentially in one direction is not only of fundamental importance, but it will enable high radiative efficiency critical for many thermal sensing, imaging, and energy devices. Over the years, different photonic materials and structures have been designed utilizing resonant and propagating modes to generate directional thermal emission. However, such thermal emission is narrowband and polarised, leading to limited thermal transfer efficiency. Here we experimentally demonstrate ultrabroadband polarisation-independent directional control of thermal radiation with a pixelated directional micro-emitter. Our compact pixelated directional micro-emitter facilitates tunable angular control of thermal radiation through non-imaging optical principles, producing a large emissivity contrast at different view angles. Using this platform, we further create a pixelated infrared display, where information is only observable at certain directions. Our pixelated non-imaging micro-optics approach can enable efficient radiative cooling, infrared spectroscopy, thermophotovoltaics, and thermal camouflaging.

Thermal radiation is a fundamental mechanism of energy transport that forms the foundation of many energy harvesting[1,2], infrared (IR) astronomy[3,4], sensing[5,6], and thermal management[7–10] devices. Thermal radiation from conventional materials creates heat transfer channels between objects and their surroundings in all directions. On the other hand, the ability to thermally couple objects to their surroundings only in certain directions (Fig. 1a), by directional thermal emission, may open the door to enhanced radiative cooling[11–14], thermal sensing[15,16], thermophotovoltaics[17,18], and incandescent infrared lighting[19]. Thus, directional thermal radiation was highly sought after[12,16,19,20] and has been realised over narrow spectral ranges with nanoscale structures such as phonon polariton gratings[20–22], plasmonic metasurfaces[19,23,24], nanoantenna[25,26], and bull's eye structures[27,28]. However, controlling narrowband thermal emission disregards much of the ideal blackbody thermal emission spectrum, severely reducing radiative efficiency (Fig. 1b), while high radiative efficiency is desired and often critical for device applications. Given that the upper limit of far-field thermal radiation intensity is that of a blackbody, an ideal directional thermal radiator needs to exhibit polarisation independence, close-to-unity

emissivity over a narrow solid angle, and operate across a broad range of wavelengths like a blackbody.

Recent efforts have been made to increase radiative performance by removing the bandwidth limitation of directional thermal radiators. Proposals to expand the spectral domain of directional thermal radiation include utilizing the Brewster effect[11,29], impedance engineering[30,31], and gradient epsilon-near-zero (ENZ) materials[32–34]. Brewster effect photonic structures are composed of many stacks of photonic band gap materials which together filter non-normal emission over a broad spectral range[35]. However, highly directional angular control on this platform requires hundreds of layers of anisotropic materials that are also transparent in the mid-IR range[11], hindering experimental demonstration. Researchers have also engineered free space impedance matching of plasmonic modes in metallic metasurfaces to induce directionality, but Fresnel equations dictate considerable emissivity even in undesired directions[30,31]. Most recently, gradient epsilon-near-zero (ENZ) materials were experimentally realised to leverage the directional thermal selectivity of individual ENZ material layers for a wide, cone-shaped thermal emission with

[1]Department of Aerospace Engineering, Texas A&M University, College Station, USA. [2]School of Electronic Science and Technology, Eastern Institute of Technology, Ningbo, China. [3]Department of Materials Science and Engineering, Texas A&M University, College Station, USA. ✉e-mail: zijing@tamu.edu

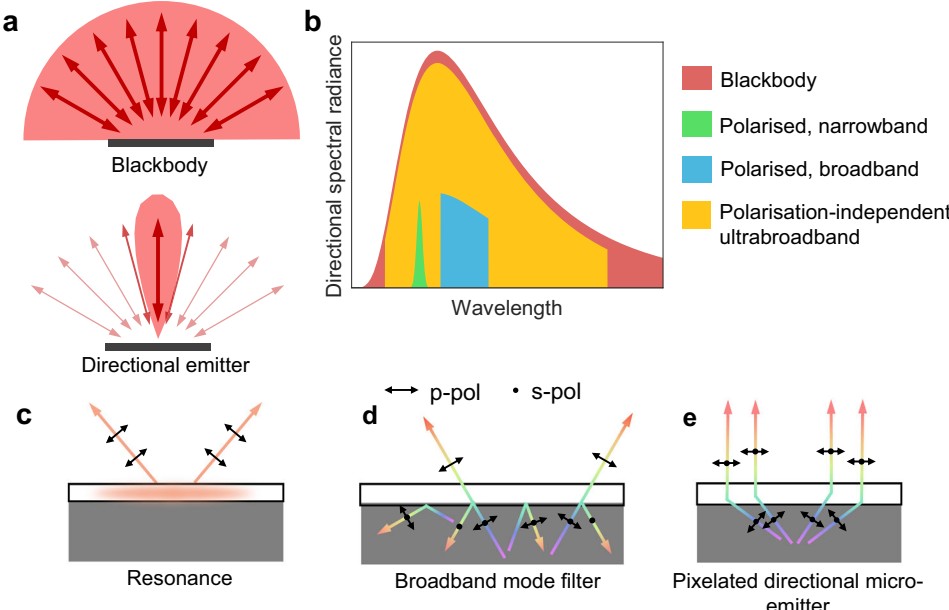

**Fig. 1 | Polarisation-independent broadband directional control of thermal emission. a** Schematic of surfaces with isotropic and directional emissivity. Whereas a blackbody exhibits uniform radiative coupling to its surroundings, radiative coupling for a directional emitter is selective for a specific direction. **b** The theoretical spectral irradiance of blackbody emission, polarised broad- and narrowband thermal emission as well as ultra-broadband polarisation-independent thermal emission. Only the last matches blackbody emission to achieve high radiance efficiency while the other two attain only a small fraction of blackbody radiance. **c** Directionality obtained via resonant approaches exhibits polarised emission over a narrow spectral range. **d** Angular filters based on propagating modes operate over a broad wavelength range but inherit polarisation dependence and produce lower radiance efficiency. **e** A pixelated directional micro-emitter based on etendue conservation gives rise to broadband directional control of both s- and p-polarised thermal radiation.

moderate emissivity. ENZ thin films rely on Berreman resonances, a type of p-polarised phonon polariton resonance with dispersive emissivity, which leads to reduced directionality[32–34].

In addition, all the above-mentioned concepts produce thermal radiation that inherits strict polarisation dependence. Thermal radiation arising from an electromagnetic resonance (Fig. 1c) is polarised corresponding to the symmetry of the resonant fields[20,27,34]. Similarly, propagating modes in anisotropic media have a characteristic polarisation, and infrared filters based on the selective transmittance of such modes[11,30,31] are generally polarised (Fig. 1d). These polarisation dependences limit radiative efficiency to half of what is achievable by an ideal directional thermal radiator, which is polarisation-independent (Fig. 1e). Thus, to achieve highly efficient, directional radiative transfer, a new class of structures based on non-resonant and non-modal design rules is desired. Such thermal surfaces will unlock the long-sought goal of achieving ultra-broadband and polarisation-independent directional thermal radiation.

One of the most exciting applications of thermal radiation is thermal camouflage, which conceals a body from infrared sensors through thermal emission engineering. Beyond traditional methods that utilise low-emissivity materials for concealment[36,37], recent technologies explored different strategies to control the spatial distribution of thermal radiation and display complex infrared information[38–40]. However, current research in this field has yet to leverage directionality in thermal camouflage. The ability to thermally emit different signals at different angles provides an extra degree of freedom to channel signals, enabling a secured and information-rich IR display, which motivates our work.

In this work, we create a compact pixelated directional micro-emitter that is broadband, polarisation independent and demonstrates strong directionality in thermal emission (Fig. 1e). Through delicate structure designing, each pixel of our directional micro-emitter leverages etendue conservation[41–43] to restrict thermal radiation within a narrow angular range. While the conservation of etendue is

exploited in the field of non-imaging optics to build large macroscopic light concentrators and collimators[44–49], its potential for microscopic thermal radiation and high-resolution thermal display has rarely been explored. Existing works with non-imaging microstructures primarily focus on control of visible light[50–57], which is fundamentally different from thermally driven infrared emission. Moreover, these structures are often bulky[50], have compromised directionality[52,53] and employ isolated microstructures[54,55] which hinder macroscopic engineering of electromagnetic properties. For thermal radiation manipulation, the few existing works propose one-dimensional slant-wall grooves, whose directionality lacks sharp angular cut-offs and full-azimuthal control, and only thick millimetre-sized grooves with limited directionality have been achieved[48,58,59]. On the contrary, our pixelated directional micro-emitter leverages non-imaging optical principles to enable efficient beam shaping while retaining a thin, compact, and integrated platform. Moreover, the inherent non-resonant operation allows for wavelength- and polarisation-independent functionality. We further exploit pixel-level control of our compact directional micro-emitter to create an infrared display that carries direction-encoded IR information and performs thermal camouflage.

## Results

We designed a three-dimensional (3D) pixelated directional micro-emitter (PDME) whose pixel structure (Fig. 2a) is composed of parabolic reflectors with an underlying SU8-coated blackbody (Supplementary Fig. 1). Our design centres on the concept of etendue conservation, where etendue is the quantity describing the product of the areal spread of light and its angular spread. By mapping thermal emission from a small area $A_0$ to a large area $A_1$ on the PDME upper surface (Fig. 2a), the angular range of thermal emission will be suppressed at $A_1$ owing to the conservation of etendue. To this end, each micro parabolic reflector was carefully positioned at the focus of its oppositely facing reflector to form a widening channel for infrared light. Furthermore, we leverage the edge ray principle to confirm the

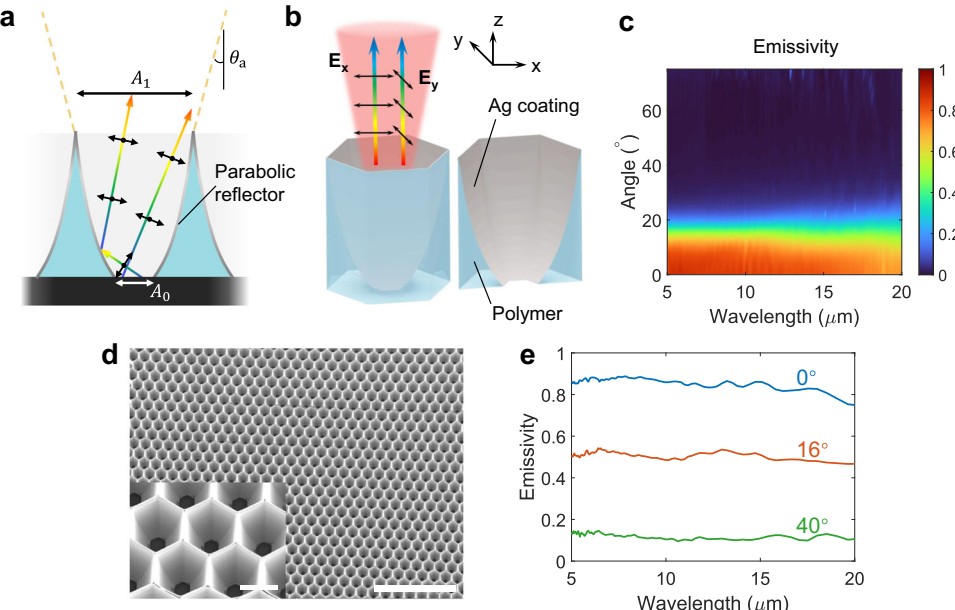

**Fig. 2 | Structure and angular-resolved spectral emissivity of a pixelated directional micro-emitter. a** 2D schematic of the pixelated directional micro-emitter (PDME) structure. Through conservation of etendue, the angular distribution of thermal radiation is compressed by enlarging the areal spread from $A_0$ to $A_1$. The control of thermal radiation is broadband and polarisation-independent. $\theta_a$ represents the acceptance angle of the PDME. **b** A 3D hexagonal pixel structure composed of three pairs of oppositely-facing parabolic reflectors is used to tessellate over a horizontal plane and minimise azimuthal anisotropy. Unpolarised thermal radiation exits the top aperture within a small angular range. $\mathbf{E_x}$ and $\mathbf{E_y}$ stand for electric field vectors, namely polarisations, along $x$ and $y$ directions. **c** Simulated angular-resolved spectral emissivity of a PDME with a 15° acceptance angle (15°-PDME), where an abrupt drop in emissivity is apparent above 15°. **d** An SEM image of a 15°-PDME, scale bar: 500 μm. The inset shows a close-up image of its 3D hexagonal pixels, scale bar: 50 μm. **e** Measured spectral emissivity of 15°-PDME showing strong directional selectivity over an ultrawide wavelength range from 5 μm to 20 μm.

mapping of all thermal emissions into a designed angular range for high thermal irradiance, which guarantees a sharp angular cutoff. Based on geometric optics, the thermal emission from the bottom aperture is therefore collimated by the parabolic reflectors. A more detailed discussion on the PDME's operation principle is provided in Supplementary Information, Section 2. In this case, our PDME strictly confines random thermal radiation emitted by SU-8-coated blackbody into the angle range of $\pm\theta_a$, where $\theta_a$ is the acceptance angle.

To tessellate over a two-dimensional (2D) plane while minimising azimuthal anisotropy, we designed our PDME pixel structures with hexagonal symmetry (Fig. 2b). Within each pixel, three pairs of micro parabolic reflectors were tailored to attain $\theta_a = 15°$, respectively. Supplementary Information Section 3 provides an in-depth introduction to the optimisation process of PDME's structural parameters. Finite element analysis results averaged over s- and p-polarisations (Fig. 2c) show that the PDME with a 15° acceptance angle (15°-PDME) exhibits directionally selective emission over an ultrawide wavelength range from 5 to 20 μm. Moreover, the emissivity spectra of the two polarisations match each other closely, as is predicted for polarisation-independent behaviour shown in Supplementary Fig. 4.

We fabricated the 15°-PDME and characterised its spectral emissivity with Fourier-transform infrared (FTIR) spectroscopy (Supplementary Fig. 5). Figure 2d shows a scanning electron microscope (SEM) image of a large-area 15°-PDME whose pixel width and height are 70 μm and 90 μm, respectively. The PDME structure was first defined with two-photon polymerisation 3D lithography, after which silver (Ag) was deposited on the curved surfaces of the as-created polymer structure via three separate runs of oblique-angle electron-beam evaporation at different orientations (Supplementary Fig. 6). The underlying SU-8/blackbody was kept metal-free by self-shadowing, and a clear bottom aperture was further ensured by using Argon plasma. We leverage the anisotropic directionality of the plasma etching process to selectively remove trace amounts of Ag at the bottom aperture

without significantly etching the Ag coated on the sidewall. The inset SEM image of Fig. 2d shows conformally Ag-coated micro parabolic reflectors with a metal-free bottom thermal emitter. The emissivity spectra measured with FTIR demonstrate strong dependence on viewing angle, with the average emissivity over 5–20 μm dropping abruptly from > 0.8 at 0° to around 0.1 at 40° (Fig. 2e). More highly resolved emissivity spectra with smaller angular intervals are shown in Supplementary Fig. 7. Moreover, these emissivity spectra exhibit little variation over different wavelengths, showing the capability for uniform directional control over an ultrabroad wavelength range. Our simulations actually predicted a theoretical working wavelength range of 0.38 to 40 μm (Supplementary Information Sections 8 and 9, Supplementary Figs. 8–10), but our measured spectral range was limited by the detection range of our equipment.

To verify PDME's near blackbody-to-mirror switching capability at different angles, we measured its directional radiative properties by thermography. To this end, our 15°-PDME was heated and rotated to collect angled thermal images with a calibrated high-resolution IR camera (Supplementary Fig. 11a). We attached a reference blackbody (emissivity shown in Supplementary Fig. 12) emitter beside the 15°-PDME to show the clear contrast between directional and quasi-isotropic thermal radiation (Fig. 3a). The recorded temperature of the 15°-PDME decreased abruptly when the angle increased from 0° to 30°, indicating a corresponding rapid drop in emissivity. In contrast, the reference quasi-isotropic emitter maintained a high emissivity at all angles. On the left-hand side of Fig. 3c, we extract average emissivity in the IR camera spectral range (7.5–14 μm) from both thermal images and simulations. The observed angular range of ±16°, over which the measured emissivity is at least half of the maximum value, closely matches the designed $\theta_a$ of 15°. Figure 3c also shows a high maximal emissivity exceeding 0.8 at 0° angle, i.e., the intended emission direction normal to the surface. Together, these results produce a clear demonstration of ultra-broadband directional thermal radiation.

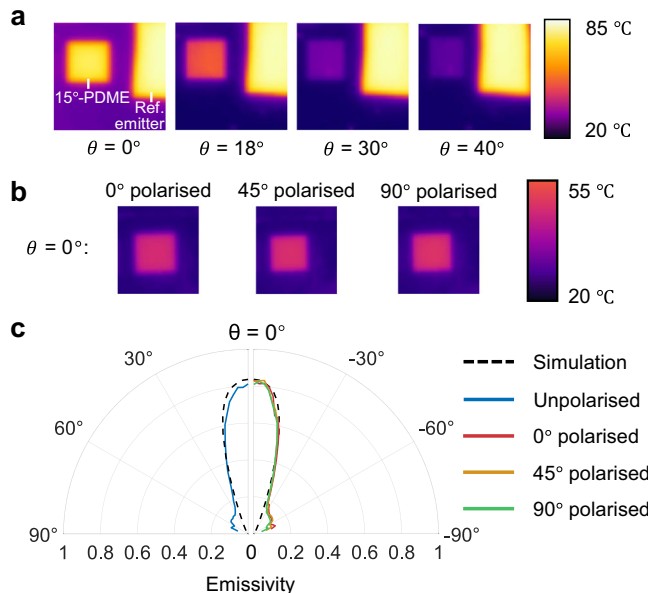

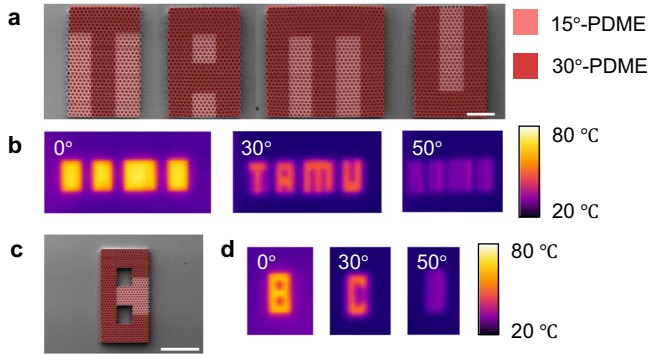

**Fig. 3 | Pixelated directional micro-emitter behaviour under thermographic imaging. a** Thermal images of an approximately 2 mm × 2 mm 15°-PDME. The structure's apparent temperature drops abruptly when the viewing angle $\theta$ increases, in contrast to a nearby quasi-isotropic reference (ref.) emitter, which maintains a constant temperature reading. **b** Thermal images for $\theta = 0°$ taken at different polarisations. The apparent temperatures are almost identical, indicating the polarisation independence of 15°-PDME (more thermal images are shown in Supplementary Fig. 13). **c** Left half of the polar plot: Average emissivity (from 7.5 to 14 μm) of the 15°-PDME extracted from thermal images. The experimental results match well with simulations, showing maximal average emissivity > 0.8 and an angular range of ±16°. Right half of the polar plot: Average emissivity (from 7.5 to 14 μm) of 15°-PDME measured with thermal radiation at different polarisations overlap with each other.

**Fig. 4 | Pixelated direction-encoded infrared displays. a**, **b** SEM image (scale bar: 500 μm) (**a**) and thermal images (**b**) of a pixelated PDME display composed of 30°-PDME pixels doped with 15°-PDME pixels. Due to the different emission angular range of the pixels, the information "TAMU" is only observable at 30° and it is camouflaged at 0° and 50°. **c**, **d** SEM image (scale bar: 1 mm) (**c**) and thermal images (**d**) of another pixelated PDME display, where true information "C" is only observable at around 30°, while misleading information such as "8" is observed at 0° and 50°.

For other photonic structures with far off-normal centre emission angles, such as gradient ENZ films[32–34] and impedance matching structures[30,31], thermal emission takes the shape of large rings in Fourier space. In contrast, the 15°-PDME thermal emission centres in the normal direction, occupying a single spot with narrow angular range. This fundamental difference in the structure of thermal emission gives rise to unprecedented directional selectivity for broadband thermal emission with compact structures.

We further demonstrated the polarisation-independent emission of our 15°-PDME by performing polarisation-filtered thermal imaging, where a linear polariser was inserted between the sample and the IR camera (Supplementary Fig. 11b). Figure 3b reveals that with a fixed viewing angle but varying polarisation, thermal images of the 15°-PDME show indistinguishable temperature readings. More detailed thermal images with different viewing angles and polarisations are shown in Supplementary Fig. 13. The presence of the polariser decreased the intensity of thermal radiation captured by the camera, so the maximal temperature reading on the 15°-PDME did not exceed 55 °C even as the sample was heated to a higher temperature. The angle-resolved average emissivities for 0°, 45° and 90° polarisations are plotted in the right half of Fig. 3c and they match closely with each other. To corroborate these thermal image results, we note that our 15°-PDME's unpolarised emissivity of > 0.8 directly implies high emissivity for both polarisations. In the case of materials that exhibit polarised thermal emission, measurements of unpolarised thermal emissivity are physically limited to 0.5.

Next, we explored the tunability of the PDME emission beam width. Through structural design, the acceptance angle of PDMEs can be arbitrarily adjusted. As a demonstration of this tuning capability, we

designed and fabricated a narrow-angle PDME (Supplementary Information Section 12 and Supplementary Fig. 14), where we measured the acceptance angle and maximal emissivity to be 11° and 0.76, respectively, for the spectral range from 7.5 to 14 μm (Supplementary Fig. 15). The broad spectral range and polarisation-independence of narrow-angle PDME were also confirmed with polarised thermal images (Supplementary Fig. 16). This verifies that PDMEs constructed for different directionality profiles maintain an ultrabroad spectral range, polarisation-independence and high directionality. For a complete demonstration of acceptance angle tuning from 11° to 30°, we also fabricated a 30°-PDME (Fig. 4a).

To further exploit the tunability of the directional micro-emitter, we devised a pixelated display for directional encoding of IR information and thermal camouflage. By doping a PDME with pixels of different acceptance angles, we created IR information that was directionally camouflaged to its surroundings. In other words, we designed messages such that they became observable only from a specific range of directions. Figure 4a shows the SEM image for a patterned 30°-PDME doped by 15°-PDME pixels. A pattern showing "TAMU" was observed only near 30°, while at other angles such as 0° and 50°, the letters were hidden and only "■ ■ ■ ■" was shown (Fig. 4b). The total area of the patterns reached centimetre scale, demonstrating PDME's potential for scalability. In addition to thermal camouflage, we also demonstrated a doped PDME's ability to directionally encode true information with misleading information within the same micro-emitter. Such a hybrid PDME was fabricated (Fig. 4c) and its direction-encoded IR pattern is shown in Fig. 4d. Here, a letter C could be observed from near 30° while a number 8 was observed at 0° and 50°. Finer sweep of the viewing angles over a larger range can be found in Supplementary Movie 1, which again confirms that information can be camouflaged and falsely displayed at the desired angle.

In summary, we created a compact pixelated directional micro-emitter to realise ultra-broadband polarisation-independent directional thermal radiation. The PDME exhibited performance superior to previous directional thermal emitters, including polarisation independence, high emissivity (> 0.8), narrow angular range (down to ±11°) and a measurement-limited ultrawide working wavelength range (5–20 μm). Therefore, the PDME demonstrates the properties of a highly directional thermal radiator and enables efficient thermal energy transfer. We further demonstrated the ability to encode information into the viewing angle of infrared displays by engineering the

emission angle range of individual PDME pixels. In this way, IR information was camouflaged or replaced with misleading information when observed from the designed range of directions. Moreover, the PDME has the benefits of being thin, scalable, and fabricable on a compact integrated platform. The non-imaging approach to directional thermal radiation achieved here will enhance the capabilities of radiative cooling, thermophotovoltaics and sensing devices, expand the scope of secured communication, and pave the road for high-capacity IR camouflage and cryptography.

## Methods
### Simulation
All simulations in this work were performed with a finite-element-method solver in COMSOL Multiphysics. Ray optics simulation was conducted for a two-dimensional scheme (Supplementary Fig. 2a) where the parabolic reflectors were set to be mirrors and the boundary condition "release from boundary" was selected for the top aperture. For wave optics simulation, periodic conditions were applied on the edges of the hexagonal pixel, whereas the top and bottom of the simulated region were bounded by perfectly matched layers. Perfect electric conductors were used to model Ag thin films on the PDME structure.

### Fabrication
SU-8 photoresist was spin-coated on a blackbody to provide a flat surface with high mid-infrared (IR) absorptivity/emissivity. Supplementary Fig. 6 shows an illustration of the fabrication process. Polymer structures of a pixelated directional micro-emitter (PDME) were then fabricated on the SU-8 layer by two-photon polymerisation (TPP) based three-dimensional (3D) nanolithography. The galvo scan speed and laser power were 10000 $\mu$m/s and 30 mW, respectively. After the laser exposure, the polymer structures were developed in propylene glycol monomethyl ether acetate for 20 min, in isopropanol for 5.5 min and in methoxynonafluorobutane for 2.5 min. The structures were then cured by ultraviolet light.

The polymer structure was mounted on a homemade glass sample holder for oblique angle electron beam deposition of silver. A 400 nm thick silver layer was deposited with 5 Å/s deposition rate. To coat all surfaces of PDME with silver, the electron beam deposition was performed sequentially from 3 different azimuthal angles. Silver deposited on the bottom aperture was then removed by argon plasma etching. We leveraged the anisotropic directionality of the plasma etching process and carefully controlled the dry etch time to attain the desired PDME structure.

### Characterisation
**Fourier-transform infrared spectroscopy characterisation.** For the blackbody, the SU-8/blackbody bilayer and the silver film, we measured their reflectance by using a Nicolet 380 Fourier-transform infrared (FTIR) spectrometer with a variable angle reflection accessory. The reflectivity spectra were collected from different angles: from 0° to 30° with a 2° step and from 35° to 75° with a 5° step.

Directional measurement was conducted for the spectral emissivity of the PDME. The measurement setup is shown in Supplementary Fig. 5a. The PDME acts as an emitter and we compare its emission with that of a known emitter to calculate its spectral emissivity. The PDME and reference samples were heated to 102 °C and their temperatures were measured with a thermistor. Since a directional emitter only emits in a narrow angular range, its total emission is weaker than that of a same-size, same-temperature quasi-isotropic emitter. An aperture was added to the light path so only light from a narrow range of angles was collected and precise emissivity could be determined for a certain direction. Supplementary Information Section 5 shows how angular-resolved spectral emissivity for PDME is calculated with the FTIR measurement results.

**Infrared image collection and analysis.** The infrared images were collected with a calibrated high-resolution infrared camera (FLIR T650sc). The spectral range of this camera is 7.5–14 $\mu$m. Unpolarised average emissivity was measured with the setup shown in Supplementary Fig. 11a. A piece of blackbody was placed beside the PDME as a reference to demonstrate the narrow angular width of the PDME.

To investigate the polarisation dependence of PDME's thermal emission, a wire grid polariser (spectral range: 2.5–20 $\mu$m) was positioned between the IR camera and the PDME sample. In addition to varying the rotation angle $\theta$ of the PDME sample, we also rotated the polariser to different angle $\phi$ to collect thermographs with different polarisation directions (Supplementary Fig. 11b). Supplementary Information Section 9 shows how directional average emissivity for PDME is calculated with the thermography measurement results.

## Data availability
The data that support the results of this study are available on Figshare (https://doi.org/10.6084/m9.figshare.25637166).

## Code availability
The codes used for analyses in this study are available from the corresponding author upon request.

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

## Acknowledgements

This work was supported by the President's Excellence Fund (X-Grant).

## Author contributions

Z.F. and Z.J.W. conceived the idea and designed the experiments. Z.J.W. supervised the project. Z.F. and S.L. performed the simulations and theoretical analysis. T.H. and Y.C. fabricated the samples. Z.F. performed the measurements. Z.F., T.H., S.L., Y.C. and Z.J.W. analysed the data. Z.F. wrote the paper with inputs from all authors.

## Competing interests

The authors declare no competing interests.
