## [Peer Review File · Nature Communications]

Directional thermal emission and display using pixelated non-imaging micro-opticsREVIEWER COMMENTS

Reviewer #1 (Remarks to the Author):

This study focuses on creating a directional micro-emitter for thermal radiation that operates across a broad spectrum and is polarization-independent. The design leverages etendue conservation to confine thermal radiation within a narrow angular range, addressing challenges of polarization dependence and spectral limitations in existing approaches. The work aims to advance the field of thermal radiation control, with potential implications for thermal sensing, thermophotovoltaics, and thermal display technologies. I like this work and think it can be published in Nature Communications after minor revisions.

1- The title is too long.

2- The concept of the conservation of etendue is well known. That said, I suggest adding a useful explanation based on the geometric optic properties of parabolic reflectors, i.e., that it collimates the thermal radiation. This will make the operation principle even clearer to the broad readership of Nat. Comm.

3- I don't understand why the directional thermal emission wavelength range is limited to 5-20 microns? It seems to me that the deposited Ag should operate over a wide wavelength range. Is it because you would need to operate at very high temperatures to measure significant blackbody radiation at shorter wavelengths and you cannot do that with your structure?

4- The fabrication itself is very interesting. If this is a new fab procedure, I recommend adding the fabrication steps as a figure and in the main manuscript. If not, then please reference sources on the fab process.

5- I like the camouflage demo, however, it is not well motivated in the intro or the results section. Some motivation somewhere is needed with some discussion of the state-of-the-art.

Reviewer #2 (Remarks to the Author):

In this work, the authors reported a non-modal design scheme for directional thermal emission based on non-imaging micro-optics. A pixelated display with broadband and unpolarized emissivity was fabricated by the 3D lithography and experimentally studied by

angle-polarization-dependent infrared spectroscopy and thermal mapping. Overall, the work is very interesting and suitable for publication in Nature Communications. The following issues should be addressed:

1. Because the pivotal concept for the work is the “non-imaging micro-optics”, it is important to elaborate the design steps of the micro-emitter. The authors need to clearly mention the tools (e.g., analytical models, design software) and intermediate results (e.g., figures, optimization of parameters) in the design and optimization process.
2. Specifically, the authors should provide more discussion about the dependence of the cutoff emission angle θ_a and cutoff emission wavelength bandwidth (upper and lower bounds) on the design parameters: height, curvature, A_1 and A_0 shape, etc. of the parabolic (or other kinds) reflector. The only principle mentioned is the etendue conservation stating $A_0 < A_1$ for reducing the angle spreading θ_a . Supplementary Fig.6 provides some discussion about the upper bound of cutoff emission wavelength along $\theta=0$ based on a waveguide understanding. However, the authors need to systematically quantify these controlling factors to substantiate the contribution of non-imaging micro-optics to thermal radiation engineering.
3. As the authors were motivated by the micro-emitters with high-radiative efficiency for energy applications and pixelated display for information applications, the authors should also discuss some dependence of the radiative energy transfer efficiency and display resolution based on the current design parameters.
4. Based on the current design, the beaming of thermal emission is not arbitrarily controllable because it is always centered at $\theta=0$. Based on the principles in Q2, can this constraint be eliminated?
5. It is also critical to discuss the validity range of the design scheme based on “non-imaging micro-optics” and its scalability on the size of the micro-emitter, compared to other nanophotonic schemes.

6. The experimental demonstrations included large angle changes (close to 90°), the array of micro-emitters may not be in uniformly focused due to the limited depth of focus for both the spectrometer and thermal camera. The authors need to verify their observation robustness against this effect, in particular for experimental results based on accurate temperatures.

Responses to Reviewer #1

We thank the reviewer for his/her positive evaluation and constructive comments. We are glad that the reviewer likes our work and thinks that it can be published in Nature Communications after minor revisions. We have addressed the comments that the reviewer pointed out and revised the manuscript accordingly. In following, we provide detailed responses to each question, comment or suggestion raised by the reviewer.

Comment 1:

“The title is too long”

Reply 1:

We agree that the title is indeed too long. Now we streamline the title to “*Directional thermal emission and display using pixelated non-imaging micro-optics*” (page 1, line 1-2).

Comment 2:

“The concept of the conservation of etendue is well known. That said, I suggest adding a useful explanation based on the geometric optic properties of parabolic reflectors, i.e., that it collimates the thermal radiation. This will make the operation principle even clearer to the broad readership of Nat. Comm.”

Reply 2:

Thank you for the comment. We have added the following two sentences in the main text (page 3, line 109-111):

“Based on geometric optics, the thermal emission from the bottom aperture is therefore collimated by the parabolic reflectors. More detailed discussion on the PDME’s operation principle is provided in Section 2 of the Supplementary Information.”

In addition, a new Section 2 is added in the Supplementary Information to offer an in-depth explanation of the mechanism:

We examine an individual pixel within the pixelated directional micro-emitter (PDME). For simplicity, the pixel is presented in a two-dimensional form, and we investigate thermal radiation incident from different directions (Supplementary Fig. 2a). For normal incidence, all incident radiation is redirected to the bottom absorber. Assuming an ideal absorber, the pixel achieves unity absorptivity in this case. When the angle of incidence θ increases, but still smaller than the acceptance angle θ_a , the pixel still demonstrates unity absorptivity. When θ reaches θ_a , all incident thermal radiation is focused on the edge of the parabolic reflector. When θ exceeds θ_a , the light rays no longer reach the bottom absorber, and zero absorptivity is observed. The relationship between absorptivity and θ is summarized in Supplementary Fig. 2b. The absorptivity drops abruptly at θ_a .

Consequently, only light incident from $-\theta_a$ to $+\theta_a$ reaches the bottom aperture. According to the reversibility of light, thermal radiation from the bottom aperture will be redirected and restricted within the $-\theta_a$ to $+\theta_a$ range upon exiting the PDME from the top, verifying the collimation effect of the parabolic reflectors (Supplementary Fig. 2c).

The principles introduced in this section are based on geometric optics where the structural feature size is larger than the wavelength. This holds true for our PDME, though it has a much smaller dimension than typical geometric optical components. Our PDME thus follows the operation principle discussed in this section, demonstrating strong directionality, as well as wavelength and polarisation independence (Supplementary Fig. 4, Fig. 2e).

Supplementary Fig. 2 | Ray trace analysis of pixelated directional micro-emitter (PDME). (a) Ray traces of thermal radiation incident from different directions. A pair of parabolic reflectors are positioned on an ideal absorber. (b) The relationship between absorptivity/emissivity and angle of incidence θ . (c) The parabolic reflector collimates random thermal radiation emitted from the bottom aperture. After redirected by the parabolic reflector pair, the thermal radiation is restricted within a narrow angular range.

Comment 3:

“I dont understand why the directional thermal emission wavelength range is limited to 5- 20 microns? It seems to me that the deposited Ag should operate over a wide wavelength range. Is it because you would need to operate at very high temperatures to measure significant blackbody radiation at shorter wavelengths and you cannot do that with your structure?”

Reply 3:

Thank you for the insightful comment. We have expanded our discussion on the PDME's operational wavelength range in Section 8 of the Supplementary Information. The PDME is engineered based on the geometric optics principle, which means its minimum operational wavelength is unrestricted as long as the silver films maintain high reflectance. Experimentally measured reflectance of silver film, as shown in Supplementary Fig. 8a, indicates the minimum operation wavelength to be 0.38 μm , beyond which the reflectance of silver film exceeds 0.95. Conversely, wavelengths above 40 μm result in the mode size surpassing the bottom aperture size (illustrated in Supplementary Fig. 8b-c), causing PDME cutoff, consistent with waveguide theory predictions (Supplementary Equation S3). Hence, the PDME can operate within a 0.38-40 μm wavelength range, substantially exceeding the 5-20 μm range demonstrated in our experiments.

The limitation to the 5-20 μm range observed in our study was primarily attributed to the detection range of the instrument. When wavelength deviates from the peak wavelength, the blackbody radiation intensity decreases. Therefore, the testing instrument can only receive detectable signal within a certain range. At 100 $^{\circ}\text{C}$, according to the Wien's displacement law, the blackbody spectrum peaks at 7.8 μm . Below this wavelength, the intensity of blackbody radiation decreases sharply, whereas above 7.8 μm , the decline in intensity is more gradual. Consequently, the signal intensity outside the 5-20 μm range diminishes to levels undetectable by our instrument. As indicated by the reviewer, the lower wavelength range could be extended by rising the testing temperature. In our experiment, the lower temperature was chosen to avoid possible damage to the photoresist scaffold and oxidation of the silver film that might degrade the device performance. Furthermore, although the experimental demonstration was confined to a 5-20 μm range, this already represents a significant advancement over existing technologies^{R1-3} and suffices to validate the PDME concept. In the main text, we have written the following (page 4, line 140-143):

Our simulations actually predicted a theoretical working wavelength range of 0.38 to 40 μm (Supplementary Information Section 8 and 9, Supplementary Fig. 8-10), but our measured spectral range was limited by the detection range of our equipment.

Further details can be found in the updated Supplementary Fig. 8 and Section 8 in the Supplementary Information (presented below).

As demonstrated in Section 2, the operation principle of PDME based on geometric optics stays solid when wavelength is much smaller than the structural feature size. Therefore, the spectral range of PDME extends to near IR and even visible range until the reflectance of silver diminishes. Supplementary Fig. 8a shows the measured reflectance of a silver film. It is observed that the silver film demonstrates high reflectance > 0.95 when the wavelength exceeds $0.38 \mu\text{m}$, which is therefore the PDME's minimum operation wavelength.

Next, we explore the maximum operation wavelength of the PDME. Finite element method (FEM) simulation was run to obtain absorbance of the PDME for different angles of incidence. Emissivity is then calculated according to the Kirchhoff's law of thermal radiation, which states that the emissivity ε of a surface is equal to its absorptivity α at a given wavelength, direction, and polarisation state^{R4}. In the simulation, we consider the SU-8/blackbody emitter as an opaque perfect absorber. Therefore, the transmission through the PDME is zero, and we have:

$$\varepsilon = \alpha = 1 - R - T = 1 - R \quad (\text{S2})$$

Angular resolved spectral emissivity $\varepsilon(\lambda, \theta)$ was simulated for both polarisations, where λ and θ stands for wavelength and angle of incidence, respectively. Supplementary Fig. 4 shows that $\varepsilon(\lambda, \theta)$ for s- and p-polarisation match each other very well.

The FEM simulation (Supplementary Fig. 8b) demonstrates that the emissivity at $\theta = 0^\circ$, $\varepsilon(\lambda, 0^\circ)$, decreases slowly with the increase of wavelength but remains higher than half of the maximum (i.e. 0.84, as shown in Fig. 3c) when $\lambda < 40 \mu\text{m}$, while the emissivity drops rapidly when $\lambda > 40 \mu\text{m}$. Therefore, the PDME has the potential to work in a very broad spectral range with a cut-off wavelength of $40 \mu\text{m}$, which can be explained if we consider the unit cells of PDME as waveguides.

The cut-off wavelength of a circular waveguide can be calculated with

$$\lambda_c = \frac{2\pi r}{1.8412} \quad (\text{S3})$$

where λ_c and r represents the cut-off wavelength and waveguide radius, respectively^{R5}. If we take $r = 11.6 \mu\text{m}$, equivalent to the radius of bottom aperture's circumscribed circle, the cut-off wavelength is calculated to be $40 \mu\text{m}$. This is in good match with the simulation result (Supplementary Fig. 8b).

To further understand the cut off at $40 \mu\text{m}$, we show the electric field distribution in the PDME and bottom absorber/emitter at normal incidence but different wavelengths (Supplementary Fig. 8c). It is observed that the mode diameter d_{mode} increases with the wavelength λ . At $\lambda = 30 \mu\text{m}$, the diameter of the mode is smaller than that of the bottom aperture and the electric field propagates into the absorber with little hindrance. High absorptivity/emissivity is achieved in this case. While λ reaches $40 \mu\text{m}$, the

mode diameter is as large as the bottom aperture diameter (d_{apt}) and an emissivity of 0.42 is obtained, half of the emissivity at 0° shown in Fig. 3c. When λ further increases and reaches $50 \mu\text{m}$, the mode diameter exceeds the bottom aperture diameter. Only a small fraction of electric field reaches the absorber and therefore the emissivity is low.

Supplementary Fig. 8 | Spectral range of PDME. **a**, Experimentally measured reflectance of a silver film. When wavelength is larger than $0.38 \mu\text{m}$, high reflectance > 0.95 is reached. Therefore, the minimum operation wavelength of the PDME is $0.38 \mu\text{m}$. **b**, Simulation result shows that the PDME cuts off at $40 \mu\text{m}$. Therefore, 15° -PDME has the potential to demonstrate directional thermal emission over a spectral range of $0.38\text{-}40 \mu\text{m}$, much wider than the experimentally measured $5\text{-}20 \mu\text{m}$ range. **c**, Simulated electric field distribution on the cross section of 15° -PDME for different wavelength $\lambda = 30, 40$ and $50 \mu\text{m}$ at normal incidence. When wavelength increases, the mode diameter d_{mode} increases and exceeds the bottom aperture diameter d_{apt} , which leads to an increased reflection, causing the absorptivity/emissivity of PDME to drop.

Comment 4:

“The fabrication itself is very interesting. If this is a new fab procedure, I recommend adding the fabrication steps as a figure and in the main manuscript. If not, then please reference sources on the fab process.”

Reply 4:

We appreciate your positive feedback regarding our fabrication technique. Our

fabrication mainly consists of 3 steps: two-photon polymerization lithography^{R6,7}, oblique-angle electron beam deposition^{R8-10} and argon plasma etching of metal^{R11}. Each individual step is based on established methods, as acknowledged through citations. Furthermore, the primary focus of our paper is to elaborate the underlying mechanism and demonstrate the ultrabroadband directional thermal radiation effect. Therefore, we find it more suitable to keep the fabrication procedure in the supplementary information, despite we believe the integration of these steps represents a novel fabrication approach.

We also insert the following paragraph in Supplementary Information Section 6 to introduce the background of our fabrication process.

Our fabrication mainly consists of 3 steps: two-photon polymerization lithography^{R6,7}, oblique-angle electron beam deposition^{R8-10} and argon plasma etching of metal^{R11}. Each individual step is based on established methods, as acknowledged through citations. However, we believe the integration of these steps represents a novel approach to fabricate complex 3D metal-dielectric micro/nanostructures targeting different functionalities and applications.

To further clarify the fabrication steps, we have created an improved illustration of the fabrication process in Supplementary Figure 6, as shown below.

Supplementary Fig. 6 | Fabrication process. SU-8 photoresist is spin-coated on a blackbody, then polymer structures of PDME are fabricated by two-photon polymerisation (TPP) 3D nanolithography. Oblique-angle deposition of silver and argon plasma etching are performed to coat silver on the parabolic reflectors while keeping the bottom aperture free of metal.

Comment 5:

I like the camouflage demo, however, it is not well motivated in the intro or the results

section. Some motivation somewhere is needed with some discussion of the state-of-the-art".”

Reply 5:

Thank you for the valuable input. We have added the following paragraph in the Introduction section of the main text (page 2, line 70-77).

One of the most exciting applications in thermal radiation is thermal camouflage, which conceals a body from infrared sensors through thermal emission engineering. Beyond traditional methods that utilize low-emissivity materials for concealment^{R12,13}, recent technologies explored different strategies to control the spatial distribution of thermal radiation and display complex infrared information^{R14-16}. However, current research in this field has yet to leverage directionality in thermal camouflage. The ability to thermally emit different signals at different angles provides an extra degree of freedom to channel signals, enabling secured and information-rich IR display, which motivates our work.

Responses to Reviewer #2

We appreciate the valuable feedback and we are delighted that the reviewer find our work very interesting and suitable for publication in Nature Communications. Below are our responses to each of the question, comments and suggestions raised by the reviewer and we have revised our manuscript accordingly.

Comment 1:

“Because the pivotal concept for the work is the “non-imaging micro-optics”, it is important to elaborate the design steps of the micro-emitter. The authors need to clearly mention the tools (e.g., analytical models, design software) and intermediate results (e.g., figures, optimization of parameters) in the design and optimization process.”

Reply 1:

Thank you for the comment. The study began with a ray optics analysis (detailed in Section 2, Supplementary Information), and structural tuning was conducted via wave optics simulations. We started the wave optics simulation from a two-dimensional (2D) scheme to gain physical insights before proceeding to carry out three-dimensional (3D) simulations. The results were rigorous enough and therefore no analytical model was needed. Both ray and wave optics simulation were conducted using COMSOL Multiphysics. The simulation settings and the software used have been mentioned in the Methods section of the main text (page 5, line 224-226), which is copied as follows.

All simulations in this work were performed with a finite-element-method solver in COMSOL Multiphysics. Ray optics simulation was conducted for a two-dimensional scheme (Supplementary Fig. 2a) where the parabolic reflectors were set to be mirrors and boundary condition “release from boundary” was selected for the top aperture. In wave optics simulation, periodic conditions were applied on the edges of the hexagonal pixel, whereas the top and bottom of the simulated region were bounded by perfectly matched layers. Perfect electric conductors were used to model Ag thin films on the PDME structure.

More importantly, we have created a new section in the SI to describe the design steps of our PDME in great details. Thus, the following sentence has been added in the main text (page 3, line 117-118):

Supplementary Information Section 3 provides an in-depth discussion on the design and optimization of PDME’s structural parameters.

The added discussion in the Supplementary Information Section 3 is shown below:

The structure of a pixelated directional micro-emitter (PDME) is determined by three parameters: acceptance angle θ_a , bottom aperture width w_0 and truncation ratio h/H . The following Supplementary Fig. 3 shows how these three parameters affect the

structure of a 2D PDME.

An acceptance angle of 15° was chosen, leading to an angular range significantly narrower than previous works^{R1,3,17}. The 2D design of corresponding parabolic reflectors is shown in Supplementary Fig. 3a. For PDME with a known θ_a , its lateral dimension is mainly determined by its bottom aperture width. When w_0 decreases, every dimension decreases proportionally (Supplementary Fig. 3b). To minimize the size of PDME while maintaining its high emissivity, wave optics simulation was performed to determine the smallest w_0 which allows high average emissivity at 0° . As shown in Supplementary Fig. 3e, the emissivity averaged over 5-20 μm first increases with w_0 and then become stable. When $w_0 = 20 \mu\text{m}$, the average emissivity for the 2D scheme reaches 0.85 which is a satisfactory value, and the average emissivity does not increase rapidly with w_0 after 20 μm . Therefore, 20 μm w_0 is regarded as an optimal balance between compact size and high emissivity.

Since the top part of the parabolic reflectors are nearly vertical, it does not significantly affect the performance of the parabolic reflectors^{R18}. Therefore, we truncate the top part to enable easier fabrication without dramatically compromising the directionality of the PDME (Supplementary Fig. 3c). We take wavelength of 10 μm as an example, and the relationship between angular-resolved emissivity and truncation ratio h/H is demonstrated in Supplementary Fig. 3f. When the h/H ratio decreases, directionality of PDME slightly degrades: the cut-off edge becomes less sharp. We selected h/H to be 0.46 as it offered the best compromise between directionality and ease of fabrication.

Having characterized the 2D PDME, we proceeded with 3D simulations of a hexagonal PDME to confirm the strong directionality observed in the 2D model (Fig. 2c shows the 3D simulation result). Moreover, the 3D PDME demonstrates stronger polarisation independence than the 2D case due to its higher symmetry, as indicated by Supplementary Fig. 3g. In the 2D case, the emissivity of TE and TM polarisation shows non-negligible difference while in the 3D case, the emissivity of the two polarisations are identical.

Furthermore, we designed 8° -PDME to show the tunability of θ_a and the feasibility to achieve small θ_a values. θ_a not only controls the curvature of the PDME, but also the PDME's height, which increases when θ_a decreases. As shown in Supplementary Fig. 3d, if an 8° -PDME has the same w_0 as a 15° -PDME, the former is 3 times taller than the latter.

Supplementary Figure 3 | Designing of PDME in 2D scheme. a-d, Parabolic reflector pairs for different PDMEs. a, 15°-PDME; b, 15°-PDME with smaller w_0 ; c, Top-truncated 15°-PDME; d, 8°-PDME, exhibiting different curvature than the 15°-PDME. e, The relationship between emissivity averaged over the range of 5 to 20 μm and bottom aperture width w_0 . f, Angular-resolved emissivity at 10 μm for different truncation ratios h/H . g, Angular-resolved emissivity at 10 μm for 2D and 3D designs. The 3D design demonstrates enhanced polarisation independence due to higher symmetry.

Comment 2:

“Specifically, the authors should provide more discussion about the dependence of the cutoff emission angle θ_a and cutoff emission wavelength bandwidth (upper and lower bounds) on the design parameters: height, curvature, A_1 and A_0 shape, etc. of the parabolic (or other kinds) reflector. The only principle mentioned is the etendue conservation stating $A_0 < A_1$ for reducing the angle spreading θ_a . Supplementary Fig.6 provides some discussion about the upper bound of cutoff emission wavelength along $\theta=0$ based on a waveguide understanding. However, the authors need to systematically quantify these controlling factors to substantiate the contribution of non-imaging micro-optics to thermal radiation engineering.”

Reply 2:

We agree that a systematic discussion on the factors influencing cutoff emission angle and wavelength is necessary. The following discussion on how θ_a determines the parabolic structures can be found in Supplementary Information Section 9:

The cutoff emission angle θ_a is a fundamental parameter in our design determining

the curvature of the parabolic reflectors. When constructing the parabolic reflectors, we first consider a curve described by the parametric equations $x = 2ps$, $y = ps^2$, where $p = \frac{1}{2}w_0(\sin\theta_a + 1)$, $\frac{q\cos\theta_a + p\sin(2\theta_a)}{2p} \leq s \leq s_{max}$, $q = -0.5p\sec^2\theta_a(-4 + 5\sin\theta_a + \sin(3\theta_a))$. The right-side parabolic reflector is obtained through rotating the given curve by θ_a . Similarly, the left-side parabolic reflector is obtained through the same parametric equation, but with a different range for s ($-s_{max} \leq s \leq -\frac{q\cos\theta_a + p\sin(2\theta_a)}{2p}$) and is rotated by $-\theta_a$. Therefore, θ_a is determined by the curvatures of the parabolas and the angles by which the parabolas are rotated. s_{max} determines the height of the PDME and is chosen manually when deciding the h/H ratio. In addition, as dictated by the conservation of etendue, there is a relation between θ_a , A_I and A_0 : $\frac{A_0}{A_I} = \sin^2\theta_a$, where A_I and A_0 are the area of top and bottom aperture area, respectively.

We have provided a more comprehensive discussion in Section 8 in the Supplementary Information about the upper and lower limits of the wavelength range, which is copied as follows:

As demonstrated in Section 2, the operation principle of PDME based on geometric optics stays solid when wavelength is much smaller than the structural feature size. Therefore, the spectral range of PDME extends to near IR and even visible range until the reflectance of silver diminishes. Supplementary Fig. 8a shows the measured reflectance of a silver film. It is observed that the silver film demonstrates high reflectance > 0.95 when the wavelength exceeds $0.38 \mu\text{m}$, which is therefore the PDME's minimum operation wavelength.

Next, we explore the maximum operation wavelength of the PDME. Finite element method (FEM) simulation was run to obtain absorbance of the PDME for different angles of incidence. Emissivity is then calculated according to the Kirchhoff's law of thermal radiation, which states that the emissivity ε of a surface is equal to its absorptivity α at a given wavelength, direction, and polarisation state^{R4}. In the simulation, we consider the SU-8/blackbody emitter as an opaque perfect absorber. Therefore, the transmission through the PDME is zero, and we have:

$$\varepsilon = \alpha = 1 - R - T = 1 - R \quad (\text{S2})$$

Angular resolved spectral emissivity $\varepsilon(\lambda, \theta)$ was simulated for both polarisations, where λ and θ stands for wavelength and angle of incidence, respectively. Supplementary Fig. 4 shows that $\varepsilon(\lambda, \theta)$ for s- and p-polarisation match each other very well.

The FEM simulation (Supplementary Fig. 8b) demonstrates that the emissivity at $\theta = 0^\circ$, $\varepsilon(\lambda, 0^\circ)$, decreases slowly with the increase of wavelength but remains higher than

half of the maximum (i.e. 0.84, as shown in Fig. 3c) when $\lambda < 40 \mu\text{m}$, while the emissivity drops rapidly when $\lambda > 40 \mu\text{m}$. Therefore, the PDME has the potential to work in a very broad spectral range with a cut-off wavelength of $40 \mu\text{m}$, which can be explained if we consider the unit cells of PDME as waveguides.

The cut-off wavelength of a circular waveguide can be calculated with

$$\lambda_c = \frac{2\pi r}{1.8412} \quad (\text{S3})$$

where λ_c and r represents the cut-off wavelength and waveguide radius, respectively^{R5}. If we take $r = 11.6 \mu\text{m}$, equivalent to the radius of bottom aperture's circumscribed circle, the cut-off wavelength is calculated to be $40 \mu\text{m}$. This is in good match with the simulation result (Supplementary Fig. 8b).

To further understand the cut off at $40 \mu\text{m}$, we show the electric field distribution in the PDME and bottom absorber/emitter at normal incidence but different wavelengths (Supplementary Fig. 8c). It is observed that the mode diameter d_{mode} increases with the wavelength λ . At $\lambda = 30 \mu\text{m}$, the diameter of the mode is smaller than that of the bottom aperture and the electric field propagates into the absorber with little hindrance. High absorptivity/emissivity is achieved in this case. While λ reaches $40 \mu\text{m}$, the mode diameter is as large as the bottom aperture diameter (d_{apt}) and an emissivity of 0.42 is obtained, half of the emissivity at 0° shown in Fig. 3c. When λ further increases and reaches $50 \mu\text{m}$, the mode diameter exceeds the bottom aperture diameter. Only a small fraction of electric field reaches the absorber and therefore the emissivity is low.

Supplementary Fig. 8 | Spectral range of PDME. **a**, Experimentally measured reflectance of a silver film. When wavelength is larger than $0.38 \mu\text{m}$, high reflectance > 0.95 is reached. Therefore, the minimum operation wavelength of the PDME is $0.38 \mu\text{m}$. **b**, Simulation result shows that the PDME cuts off at $40 \mu\text{m}$. Therefore, 15° -PDME has the potential to demonstrate directional thermal emission over a spectral range of $0.38\text{-}40 \mu\text{m}$, much wider than the experimentally measured $5\text{-}20 \mu\text{m}$ range. **c**, Simulated electric field distribution on the cross section of 15° -PDME for different wavelength $\lambda = 30, 40$ and $50 \mu\text{m}$ at normal incidence. When wavelength increases, the mode diameter d_{mode} increases and exceeds the bottom aperture diameter d_{ap} , which leads to an increased reflection, causing the absorptivity/emissivity of PDME to drop.

Furthermore, we investigate into the structural parameters' influence on the cutoff wavelength and the following can be found in a new section in the Supplementary Information (Section 9):

Among the designing parameters, the maximum operation wavelength is mainly determined by the bottom aperture width w_0 . When w_0 increases, the cutoff wavelength increases proportionally (Supplementary Fig. 9a), consistent with the prediction of equation S3 in the Supplementary Information. In contrast, the acceptance angle exerts only minor influence on the cut-off wavelength (Supplementary Fig. 9b). The 45° -PDME exhibits a cut-off wavelength smaller than the 15° - and 30° -PDME because its top aperture is no longer much larger than the bottom aperture, thereby also restricting light accessing the PDME. The cutoff wavelength for PDMEs with different

truncation ratios are almost identical (Supplementary Fig. 9c).

Supplementary Fig. 9 | Relationships between cut-off wavelength and PDME designing parameters. a, Spectral emissivity for 15°-PDMEs with different w_0 . The cut-off wavelength increases proportionally with w_0 . **b**, Spectral emissivity for PDMEs with different θ_a . The cut-off wavelengths vary slightly with θ_a . **c**, Spectral emissivity for 15°-PDMEs with different truncation ratios. It is observed that the cut-off edge does not shift when the truncation ratio varies.

Comment 3:

“As the authors were motivated by the micro-emitters with high-radiative efficiency for energy applications and pixelated display for information applications, the authors should also discuss some dependence of the radiative energy transfer efficiency and display resolution based on the current design parameters.”

Reply 3:

We appreciate your insightful comment. To evaluate the energy transfer efficiency, envision a scenario where an ideal receiver with a 10 cm diameter (d) is positioned 20 cm (D) in front of the PDME, as shown in Fig. R1a. Since the PDME size is far smaller than D and d , PDME is treated as a point source in this analysis. From the geometry, we find out that only thermal emission within $[-\theta_c, +\theta_c]$ is absorbed by the receiver and $\theta_c = 14^\circ$. Also, we assume the emissivity of the PDME is 1 for $\theta \leq \theta_a$ and 0 for $\theta \geq \theta_a$ (Supplementary Fig. 2b), where θ_a is the largest allowed angle of emission at the PDME top aperture. The radiative power of the PDME can be described by^{R19}:

$$P_{rad} = \int_0^{2\pi} d\phi \int_0^{\pi/2} \varepsilon(\theta) \sin\theta \cos\theta d\theta \int I_{BB}(T, \lambda) d\lambda,$$

where I_{BB} stands for spectral irradiance of blackbody radiation, T stands for absolute temperature and λ stands for wavelength.

Therefore, the energy transfer efficiency η can be calculated with the following equation:

$$\eta = \frac{\int_0^{\theta_c} \varepsilon(\theta) \sin\theta \cos\theta d\theta}{\int_0^{\pi/2} \varepsilon(\theta) \sin\theta \cos\theta d\theta}$$

The energy transfer efficiency for ideal PDMEs with different θ_a is calculated and plotted in Fig. R1b. When $\theta_a \leq \theta_c$, $\eta = 1$; when $\theta_a > \theta_c$, η decreases gradually. We also calculated the η for the 15°-PDME with the measured $\bar{\varepsilon}(\theta)$ shown in Fig. 3c and plot it in Fig. R1b. Its position is very close to the curve for ideal η and is about 1 order of magnitude higher than the η of isotropic thermal emitters. The discrepancy between the measured $\eta(15^\circ\text{-PDME})$ and the ideal η originates from the 15°-PDME's sub-unity maximum emissivity and non-zero emissivity at angles larger than θ_a . The energy transfer efficiency for an isotropic emitter is calculated and represented in Fig. R1 by a blue dot^{R20}, closely matching our model. This proves the validity of our approach in calculating η . From the above analysis, we conclude that η is mainly dependent on θ_a and that directionality enhances η dramatically.

Fig. R1 | Energy transfer efficiency of PDMEs. **a**, Illustration of the model for analyzing the energy transfer efficiency (η) of PDMEs. **b**, Ideal η for PDMEs with different θ_a . The red spot stands for the η calculated from the 15°-PDME measurement results, which is approximately an order of magnitude higher than that of isotropic thermal emitters (blue spot).

The resolution of the thermal display is equal to the top aperture width w_l . Before truncation, w_l can be calculated with

$$\frac{w_0}{w_l} = \sin\theta_a,$$

where w_0 , w_l and θ_a stand for the PDME's bottom aperture width, top aperture width and acceptance angle, respectively. Since truncation of the PDME only removes the top part with slope close to a vertical line, w_l is not significantly changed after truncation and the above equation remains a good estimation of the pixel size. If we do not take into account the limitation of detectors (thermal cameras, for example), the display resolution can be calculated as

$$w_l(15^\circ\text{-PDME}) = \frac{20}{\sin(15^\circ)} = 77.3 \mu\text{m},$$

since w_0 for 15°-PDME is 20 μm . To be more precise, we check the truncated top aperture size in our structural design file and w_1 of 68.9 μm was observed, close to the w_1 before truncation. Furthermore, real w_1 of 68.1 μm was measured from the SEM image shown in Fig. 2d inset, well matching the design and confirming the high structural precision of our PDME. In conclusion, a resolution of 68.1 μm was experimentally achieved.

In the thermal camouflage sample, we adjusted the w_0 of the 30°-PDME to also be 68.9 μm for seamless integration of pixels with different θ_a .

Comment 4:

“Based on the current design, the beaming of thermal emission is not arbitrarily controllable because it is always centered at $\theta=0$. Based on the principles in Q2, can this constraint be eliminated?”

Reply 4:

Thank you for the question. This constraint can be eliminated by expanding our design to a more general form. The PDME’s design, as detailed in our response to Q2, utilizes symmetric parabolic reflectors created by rotating parabolic curves through angles of $+\theta_a$ and $-\theta_a$. In the following figure, we show two asymmetric parabolic reflector pairs where the two reflectors are tilted by θ_1 and θ_2 ($\theta_2 > \theta_1$) (Figs. R2a and R2b) gives rise to oblique thermal emission. Two-dimensional wave optics simulations for the asymmetric pairs are conducted at 10 μm wavelength. It is observed that both asymmetric parabolic reflector pairs give rise to off-axis thermal emission profile. In addition, the more the left reflector tilts, the more the emission profile deviates from 0°.

Fig. R2 | Asymmetric parabolic reflector pairs and the corresponding simulated thermal emission profiles. a and b, Asymmetric parabolic reflector pairs where the parabolic reflectors are tilted by different angles θ_1 and θ_2 ($\theta_2 > \theta_1$). **c and d,** Thermal emission profiles simulated for the asymmetric parabolic reflector pairs ($\lambda = 10 \mu\text{m}$) shown in **a** and **b**, respectively. As the tilting angle of the left reflector increases, the thermal emission profile becomes more off-axis.

Comment 5:

“It is also critical to discuss the validity range of the design scheme based on “non-imaging micro-optics” and its scalability on the size of the micro-emitter, compared to other nanophotonic schemes.”

Reply 5:

Thank you for the valuable input. We discuss the validity range of wavelength in our Reply 2, as well as in Supplementary Information Section 8 (with its summarizing paragraph shown below):

As demonstrated in Supplementary Fig. 8, for our 15°-PDME, the minimum operation wavelength is 0.38 μm (Supplementary Fig. 8a), which is determined by the intrinsic property of silver, whereas the maximum operation wavelength is 40 μm (Supplementary Fig. 8b), consistent with the prediction of equation S3. The bandwidth of our PDME is at least an order of magnitude larger than other directional thermal emitters^{R1-3}.

Moreover, we discuss the validity range of the PDME’s dimension and its scalability in Supplementary Information Section 9. The related paragraphs and figure are presented here:

For a PDME, its smallest size is dependent on the maximum wavelength of interest. As an example, assume a new design scenario where the maximum wavelength of interest (or equivalently the cut-off wavelength) is 20 μm , equation S3 introduced from waveguide theory (Supplementary Information Section 8) can be transformed to:

$$\lambda_c = \frac{2\pi r}{1.8412} = \frac{2\pi}{1.8412} \frac{1}{\sqrt{3}} w_0 = 1.9702 w_0$$

, which estimates the bottom aperture width w_0 to be 10.2 μm . 3D simulation was also conducted for a 15°-PDME at 20 μm wavelength and the result is presented in Supplementary Fig. 10. It is shown that emissivity at 20 μm increases with w_0 until 11 μm , and then the emissivity almost does not change when w_0 further increases. Therefore, the smallest acceptable w_0 is found to be around 9 μm : the smallest w_0 possessing emissivity higher than half of that at $w_0 = 11 \mu\text{m}$. This simulation result closely matches the waveguide theory’s estimation (10.2 μm) from the above equation. Therefore, for a certain desired wavelength range $[0.38, \lambda_{max}]$ (μm), the smallest w_0 can be roughly estimated with $w_0 = \frac{1}{2} \lambda_{max}$. On the other hand, since the structure is

designed based on ray optics, there is no upper limit on w_0 . When the λ_{max} is large, e.g.: 40 μm , the PDME's size is larger than other nanophotonic schemes. However, if λ_{max} is small, e.g.: 2 μm , the minimum w_0 can be as small as 1 μm , comparable to or smaller than the feature size of typical nanophotonic directional thermal emitters^{R1-3}.

Supplementary Fig. 10 | The relationship between emissivity at 20 μm and w_0 , obtained with a 3D wave optics simulation.

We add the following paragraph in Supplementary Information Section 6 to discuss the possibility of large-scale fabrication of the PDME:

Our 15°-PDME possesses a periodic structure with a relatively large feature size, making it amenable to large-scale fabrication through microstereolithography and nanoimprinting. The structure's scalability can be further enhanced by increasing the feature size to accommodate low-cost commercial 3D printing and laser cutting, although this adjustment comes with trade-offs including increased thickness and larger pixel dimensions. The feature size can be chosen with flexibility to meet the requirement of both compactness and scalability, depending on the application. The scalability of the 15°-PDME reported in the manuscript is close to other directional thermal emitters with patterns^{R2,3}, but it is less scalable than the multilayer film design^{R1}. However, if the PDME is enlarged and becomes compatible with commercial 3D printing, its scalability can be significantly improved.

Comment 6:

“The experimental demonstrations included large angle changes (close to 90°), the array of micro-emitters may not be in uniformly focused due to the limited depth of focus for both the spectrometer and thermal camera. The authors need to verify their observation robustness against this effect, in particular for experimental results based on accurate temperatures.”

Reply 6:

Thank you for the insightful comment.

To prove the robustness of the thermal images captured, we compare the sample's deviation from focal plane with the measurement systems' depth of fields. Depth of field is the distance between the farthest and nearest object that can be clearly captured in an image, for which we have direct experimental evidence. The 15°- and 8°-PDME samples are approximately 2×2 mm in size. Therefore, we estimate the deviation from focus to be $0.5 \times 2 \times \sin 90^\circ = 1 \text{ mm}$. As a comparison, our thermal camera (FLIR T650sc) has a depth of field much larger than a few millimeters. As an example, in Fig. R3a, we show a thermal image of the camouflage sample taken from 60°. It is observed that most part of the sample holder is clearly imaged other than the left end. Also, the alligator clips behind the sample holder can be observed clearly in the image. From the dimensions of the sample and the sample holder, along with the rotation angle, we estimate the depth of field to be at least 26 mm.

Moreover, we measure the temperature along the white line on the Ag film region (see Fig. R3a). The maximum temperature reading on the line is 24.9 °C and the minimum is 24.6 °C, showing consistent results despite the large change in depth. Therefore, we conclude that our thermal camera has a depth of field much larger than the PDME's possible deviation from focus.

To prove the robustness of the FTIR measurement, we consider PDME's maximum deviation from focal spot not to exceed 1 mm, half the width of the PDME array. We collected spectra for the Fourier-transform IR spectrometer (FTIR) thermal source when it was placed on the focal plane and when it was 1 mm off the focal plane. Relative deviation δ between the two spectra is demonstrated in Fig. R3b. $\delta \leq 0.05$ (i.e. below 5% deviation) is observed over the measured spectral range, indicating reliable measurement results even for the largest rotation angle.

Furthermore, we obtained the emissivity of PDME by comparing the angular-resolved emission spectra of the PDME with those from reference samples. The PDME and reference samples were rotated by the same angle and thus experienced the same deviation from the focal plane of the FTIR. Therefore, even though the relative deviation caused by off-focus effects is already small, it will be further eliminated when calculating the spectral emissivity with equation S1 (Supplementary Information, Section 5).

Fig. R3. Depth of focus of characterization tools. **a**, Thermal image showing a clear image over a large range of depths. The green and red arrows indicate the position of max (24.9 °C) and min (24.6 °C) temperature on the line. **b**, Relative deviation between the emission spectra measured when the thermal source is positioned on the focal plane and 1 mm from the focal plane. The relative deviation is less than 5% over the measured spectral range.

References

- R1 Xu, J., Mandal, J. & Raman, A. P. Broadband directional control of thermal emission. *Science* **372**, 393-397 (2021).
- R2 Greffet, J.-J. *et al.* Coherent emission of light by thermal sources. *Nature* **416**, 61-64 (2002).
- R3 Costantini, D. *et al.* Plasmonic Metasurface for Directional and Frequency-Selective Thermal Emission. *Physical Review Applied* **4**, 014023 (2015).
- R4 Howell, J. R., Mengüç, M. P., Daun, K. & Siegel, R. *Thermal Radiation Heat Transfer*. (CRC press, 2020).
- R5 Balanis, C. A. *Advanced Engineering Electromagnetics*. (John Wiley & Sons, 2012).
- R6 Balli, F., Sultan, M., Lami, S. K. & Hastings, J. T. A hybrid achromatic metalens. *Nature Communications* **11**, 3892 (2020).
- R7 Ren, H. *et al.* An achromatic metafiber for focusing and imaging across the entire telecommunication range. *Nature Communications* **13**, 4183 (2022).
- R8 Barranco, A., Borrás, A., Gonzalez-Elipé, A. R. & Palmero, A. Perspectives on oblique angle deposition of thin films: From fundamentals to devices. *Progress in Materials Science* **76**, 59-153 (2016).
- R9 Karabacak, T., Wang, G. C. & Lu, T. M. Quasi-periodic nanostructures grown by oblique angle deposition. *Journal of Applied Physics* **94**, 7723-7728 (2003).
- R10 Gong, J. *Novel daylighting system based on advanced embedded optical microstructures for various facade orientation and climates*, EPFL.
- R11 Hirano, M., Hashimoto, M., Miura, K. & Ohtsu, N. Fabrication of antibacterial nanopillar surface on AISI 316 stainless steel through argon plasma etching with direct current discharge. *Surface and Coatings Technology* **406**, 126680 (2021).
- R12 Miao, L. *et al.* Design and Preparation of a Functional One-Dimensional

- Photonic Crystal with Low Emissivity in the Region of 8–14 μm . *Journal of Applied Spectroscopy* **83**, 1102-1106 (2017).
- R13 Tagliabue, G., Eghlidi, H. & Poulikakos, D. Rapid-Response Low Infrared Emission Broadband Ultrathin Plasmonic Light Absorber. *Scientific Reports* **4**, 7181 (2014).
- R14 Chandra, S., Franklin, D., Cozart, J., Safaei, A. & Chanda, D. Adaptive Multispectral Infrared Camouflage. *ACS Photonics* **5**, 4513-4519 (2018).
- R15 Li, M., Liu, D., Cheng, H., Peng, L. & Zu, M. Manipulating metals for adaptive thermal camouflage. *Science Advances* **6**, eaba3494 (2020).
- R16 Qu, Y. *et al.* Thermal camouflage based on the phase-changing material GST. *Light: Science & Applications* **7**, 26 (2018).
- R17 Kelley, K. P. *et al.* Multiple Epsilon-Near-Zero Resonances in Multilayered Cadmium Oxide: Designing Metamaterial-Like Optical Properties in Monolithic Materials. *ACS Photonics* **6**, 1139-1145 (2019).
- R18 Xiao, L., Zheng, C., Shi, K. & Chen, F. Model construction and performance research of the optimized compound parabolic concentrator based on critical truncation and multi-section congruent. *Renewable Energy* **217**, 119201 (2023).
- R19 Zhu, L., Raman, A. P. & Fan, S. Radiative cooling of solar absorbers using a visibly transparent photonic crystal thermal blackbody. *Proceedings of the National Academy of Sciences* **112**, 12282-12287 (2015).
- R20 Chung, B. T. F. & Sumitra, P. S. Radiation Shape Factors from Plane Point Sources. *Journal of Heat Transfer* **94**, 328-330 (1972).

REVIEWERS' COMMENTS

Reviewer #1 (Remarks to the Author):

The authors did a great job addressing my comments and concerns. I recommend publishing the paper without any further revisions.

Reviewer #2 (Remarks to the Author):

The authors have fully addressed my comments. I therefore recommend to publish the revised manuscript in Nature Communications.